# The Efficiency of Microstrainers Filtration in the Process of Removing Phytoplankton with Special Consideration of Cyanobacteria

**DOI:** 10.3390/toxins11050285

**Published:** 2019-05-21

**Authors:** Wanda Czyżewska, Marlena Piontek

**Affiliations:** 1Water and Sewage Laboratory, Water and Wastewater Treatment Plant in Zielona Góra, Poland, Zjednoczenia 110A, 65-120 Zielona Góra, Poland; wanda.czyzewska@zwik.zgora.pl; 2Institute of Environmental Engineering, University of Zielona Góra, Licealna 9, 65-417 Zielona Góra, Poland

**Keywords:** cyanobacteria removal, microcystins, microstrainers, particle filtration, water treatment

## Abstract

The research presented in this manuscript concerns the evaluation of the effectiveness of microstrainers, which are designed to reduce the amount of plankton in treated surface water. The efficiency of microstrainer filtration analysis is very important for the proper course of the water-treatment process not only in the Water-Treatment Plant (WTP) in Zielona Góra (central western Poland) but also in other WTPs around the world. The qualitative and quantitative monitoring of the abundance of plankton including cyanobacteria during the particle-filtration process allows not only for the assessment of the potential cyanotoxic risk in surface water providing a source of drinking water, but also allows the evaluation of the action and the prevention of adverse impacts of microstrainers. Over four years of research, it was observed that the largest amount of cyanobacteria before microstrainer filtration took place in May. The dominant species was *Limnothrix redeckei*. The microstrainer removal of plankton and cyanobacteria was statistically significant. The quantity of removed plankton increased with its increasing content in raw water. The particle-filtration process, by reducing the amount of cyanobacteria, contributes to a decrease in intracellular microcystins.

## 1. Introduction

### 1.1. The Significance of the Presence of Phytoplankton Including Cyanobacteria in the Process of Water Treatment

Pollution and eutrophication lead to the presence of high concentrations of organic and inorganic compounds which enhance phytoplankton (including cyanobacteria) blooms and concomitantly decrease water quality. The occurrence of these blooms in the source water for drinking-water production is of critical importance to drinking-water providers. Phytoplankton can have both a physical impact (e.g., clogging of filters) and chemical impact (e.g., production of cyanotoxins, disinfection by-products, and taste and odor compounds) on the treatment process [1,2,3].

Strong growth of undesirable cyanobacteria and algae species is a serious problem in reservoirs intended as drinking-water intakes for large urban agglomerations in Poland. This is the kind of problem that is faced by the Warsaw water-supply company, which delivers water from the Vistula River via the Czerniakowski Settling Pond, the dam reservoir in Przeczyce and the Zegrzyński and Dobromierz Reservoir [4]. Toxic cyanobacterial blooms occurred in the following reservoirs: Kozłowa Góra [5] Goczałkowice, Turawa, Otmuchów, and Nysa, which supply the inhabitants of Silesia with water [6] in the Dobczyce Reservoir, supplying the city of Cracow with drinking water [7] and the Sulejów Reservoir, which is the source of drinking water for the inhabitants of Łódź [4].

One of the sources of raw water for the Zielona Góra water-supply company is the water from the Obrzyca River, in whose catchment area massive blooms of cyanobacteria occur [8]. In the area under study species of toxic cyanobacteria were identified, including those which synthesize the most dangerous hepatotoxin, microcystin MC-LR [9]. Periodic cyanobacterial blooms in the catchment area of the Obrzyca River and at its source, Sławskie Lake, initiated research on the possibility of controlling cyanotoxic risk [8]. The research has been conducted with the use of microstrainers in the technological line for water purification at the Water-Treatment Plant (WTP) in Zielona Góra.

### 1.2. The Methods of Cyanobacteria and Cyanotoxin Elimination in Treated Water Including Process of Water Treatment for the Inhabitants of Zielona Góra

The amount of phytoplankton, including cyanobacteria, can be reduced in the processes of coagulation and flocculation as a result of aggregation of smaller particles into larger ones. However, studies have shown that the removal rate of cells and cyanotoxins is variable, and influenced by treatment conditions and by the phytoplankton species present in the water [10]. During coagulation, certain problems may arise such as cell lysis, leading to the release of intracellular toxins [11].

Although chloamountrination and ozonation are effective in removing microcystins, powdered activated carbon is one if the major treatment barriers for the removal of cyanotoxins in WTPs. Activated carbons are unique and versatile adsorbents because of their extended surface area, microporous structure, high adsorption capacity, and economic feasibility. The introduction of powdered activated carbon during coagulation allows for the removal of cyanobacterial toxins from drinking water [12].

Ozonation in the water-treatment chain may result in algal cells lysis and increase dissolved microcystins. Effectiveness of ozonation varies depending upon the disinfectant dose and types of cyanotoxins [11].

Membrane filtration is a physical process that separates contaminants by size and charge depending on the physical/chemical characteristics of the membrane. An important point when considering filtration is the lysis of cells. Membrane filtration is a pressure-driven process which uses size exclusion, charge repulsion, adsorption, and/or diffusion processes for removal mechanisms. Generally, there are four processes that are considered feasible and more commonly used for membrane treatment in drinking-water applications: microfiltration, ultrafiltration, nanofiltration, and reverse osmosis. Two types of low-pressure membrane filtration, microfiltration, and ultrafiltration, have been shown to be effective in the removal of intact cyanobacterial cells. These membranes removed more than 98% of the cells but are unable to reject dissolved toxins [13].

During low-pressure membrane treatments of cyanobacterial cells, including microfiltration and ultrafiltration, there have reportedly been releases of intracellular compounds including cyanotoxins and compounds with an earthy/musty odor into the water, probably owing to products of cyanobacterial cell breakage being retained on the membrane [14].

Microstrainers consist of a finely woven stainless-steel wire cloth mounted on a revolving drum that is partially submerged in the water. Water enters through an open end of the drum and flows out through the screen, leaving suspended solids behind. Captured solids are washed into a hopper when they are carried up out of the water by the rotating drum. Microstrainers consist of a very fine screen used primarily to remove algae other aquatic organisms and small debris that can clog treatment plant filters [15]. Application of microstrainers as an improvement to the water purification process is used in the WTP in Stuttgart (Constance Lake) where 12 microstrainers with a mesh width of 15 µm are used.

The Obrzyca River (the right tributary of the Oder River) is the surface water which is purified in WTP Zawada and supplied as drinking water to the local population (mainly the inhabitants of Zielona Góra). Surface water is one of the main sources of raw water used for the purposes of the municipal water-supply system in Zielona Góra [2,16]. The percentage share of particular water sources in the consumption water supply for the inhabitants of Zielona Góra in the years 2011–2014 is presented in Figure 1.

The quality of water taken from the Obrzyca River falls within category 3 (water requiring highly efficient physical and chemical treatment) [17]. The poor quality of raw water supplied to WTP Zawada and the need to treat the groundwater and surface water collectively causes numerous problems in the process of water treatment. Surface water is characterized by pollution typical of eutrophic waters (high color, oxidizability, periodic blooms, and increased levels of orthophosphates). The latter are part of household and industrial sewage, but they may also come from surface runoffs from arable land treated with artificial fertilizers containing phosphorus compounds. Surface water is strained with grids and sieves. It is then pumped through a pipeline to the first element of the processing system, where it is strained in three microstrainer drums (Figure 2) fitted with a 10 μm screening mesh. The diagram of the process of surface water treatment at WTP is presented in [2].

This study aims to demonstrate that the microstrainers operating at WTP in Zielona Góra are a useful tool for the water-treatment technology intended to eliminate plankton, including cyanobacteria and their toxins. The microscopic analysis of the abundance of plankton including cyanobacteria during the microstrainer process allows not only for the assessment of potential cyanotoxic threat in supplied drinking water, but also evaluation of the action, and prevention of adverse impacts of the microstrainers.

## 2. Results

### 2.1. The Amount of Plankton Including Cyanobacteria at WTP Zawada Before and After Microstrainers

In the analyzed raw water, the largest share was diatoms, then cyanobacteria. The smallest participation in phytoplankton was green algae. The microstrainer removal efficiency of plankton and cyanobacteria, depending on the level, are presented in Table 1.

Most samples were with the lowest level of contamination. The widest range and the lowest average of removal percentage for plankton and cyanobacteria occurred when the amount was the lowest (< 10·10^3^ org.·dm^−3^). Average removal percentage > 50% took place throughout the entire research area for plankton but > 50 10^3^ org.·dm^−3^ for cyanobacteria.

The largest amount of plankton before microstraining was observed in May 2011 and was 497·10^3^ org.·dm^-3^ (Figure 3) and the dominant group was cyanobacteria.

The smallest amount of plankton before microstraining was observed on July 4, 2012 (0.96·10^3^ org.·dm^−3^). On the same day, the amount of cyanobacteria was 0.12·10^3^ org.·dm^−3^ and the reduction in plankton reached 79.2%. The maximum degree of plankton removal, totaling 93.98%, took place in June 2011 and the minimum (9.1%) in September 2013. Removal of greater than half of the total plankton was detected in 42 samples (63.6%) including cyanobacteria (60.6%), diatoms (51.5%), green algae (63.6%).

The number of removed plankton increased with its increasing content in raw water as presented in Figure 4. The student’s t-test analysis showed significant differences between the amount of plankton (t_calc_ 2.02; t_crit_ 1,95; *p* > 0.05) before and after microstraining.

### 2.2. The Quantitative and Qualitative Analysis of Cyanobacteria at WTP in Zawada Before and After Microstrainers

The amount of cyanobacteria in samples before and after microstraining fell into the following ranges respectively: 0.02–243·10^3^ org.·dm^−3^; 0.02–118·10^3^ org.·dm^−3^ (Appendix A, Table A1). The largest amount of cyanobacteria before microstraining was observed in May 2011 and the dominant species was *Limnothrix redeckei* (Meffert).

The most frequent species of cyanobacteria found on microstrainers was *Dolichospermum affinis* (Lemmermann) (28% of the analyzed samples) (Appendix A, Table A1). Other species of cyanobacteria of the same genus that were dominant in the analyses were: *Dolichospermum flos-aquae* (Lyngbye), *Dolichospermum planctonica* (Brunnthaller), *Dolichospermum spiroides* (Klebahn). The second in terms of the frequency of dominance was *Pseudoanabaena limnetica* (Lemmermann) 19 samples and *L. redeckei* 12 samples (9.09%) of 132 samples, most frequently in May or June. *Oscillatoria granulata* (N.L. Gardner) occurred in September in 10 samples. The genus *Microcystis* was dominant in 18 samples, of which *Microcystis aeruginosa* (Kützing) in 15 samples (7.6%) in summer months and *Microcystis flos-aquae* (Wittrock) in 3 (1.5%) in August. The *Planktothrix agardhii* (Anagn. & Komárek) dominated in 8 samples in September. Other species occurring in the samples included *Aphanizomenon flos-aquae* (Ralfs ex Bornet & Flahault), *Merismopedia glauca* (Kützing), *Microcystis viridis* (Lemmermann). *M. wesenbergii* (Komárek in N.V. Kondrat.), *Oscillatoria tenuis* (Agardh ex Gomont), *Snowella lacustris* (Komárek & Hindák), and *Woronichinia naegeliana* (Elenkin).

The decrease in the amount of cyanobacteria due to microstraining ranged from 5 to 93%. The maximum degree of cyanobacteria removal occurred on 27 July 2011 (Figure 5).

The number of removed cyanobacteria increased with its increasing content in raw water. The student’s t-test analysis showed significant differences between the amount of cyanobacteria (t_calc_ 2.05; t_crit_ 1,95; *p* > 0.05) before and after microstraining.

On that day, the amount of cyanobacteria fell from 29.5 to 2.06·10^3^ org.·dm^−3^. The amounts of cyanobacteria and their reduction in particular years are presented in Appendix A, Table A1, and Figure 5.

The maximum amount of cyanobacteria in 2011, totaling 243·10^3^ org.·dm^−3^, was observed in May and the dominant species was *L. redeckei*. In 2007, as a result of the microstraining process, in 10 of the analyzed samples a decrease in the quantity of cyanobacteria by more than half was observed, which constitutes 48% of all samples examined in that year (Appendix A, Table A1).

The largest amount of cyanobacteria in 2012, totaling 70.7·10^3^ org.·dm^−3^, was detected in September, and the dominant species was *P. agardhii*. In 2012, as a result of the microstraining process, in 9 of the analyzed samples a decrease in the amount of cyanobacteria by more than half was observed, which constitutes 75% of all samples examined in that year (Appendix A, Table A1).

The maximum quantity of cyanobacteria in 2013 was in August (86.6·10^3^ org.·dm^−3^) and the dominant species were *D. spiroides, M. aeruginosa*. In 2013, as a result of microstrainer filtration, in 6 of the analyzed samples a decrease in the amount of cyanobacteria by more than half was observed, which constitutes 40% of all samples examined in that year (Appendix A, Table A1).

The largest quantity of cyanobacteria in 2014 was detected in September (213·10^3^ szt.·dm^−3^) and the dominant species was *P. agardhii*. In 2014, as a result of microstrainer work, in 5 of the analyzed samples, a decrease in the amount of cyanobacteria by more than half was observed, which constitutes 38% of all samples examined in that year (Appendix A, Table A1).

In the years 2011–2014 a decrease in the quantity of cyanobacteria by over 50% was observed in 30 samples (29.2%).

### 2.3. The Content of Microcystins and the Amount of Cyanobacteria at WTP in Zielona Gora Before and After Microstrainers

Due to the large amount of cyanobacteria in May 2011 at WTP Zielona Góra, an additional series of hydrobiological analyses was performed with a simultaneous microcystin determination using the HPLC method. In the analyzed samples, neither extracellular nor intracellular MC-YR and MC-RR microcystins were found. The only intracellular microcystin detected was MC-LR, whose content is presented in Table 2. The highest concentration of the intracellular MC-LR (0.116 µg·dm^−3^) was observed on May 11, 2011, when the amount of cyanobacteria equaled 238·10^3^ org.·dm^−3^ and the dominant species was *Limnothrix redeckei* (Appendix A, Table A1).

The MC-LR content in the cells of the cyanobacteria sampled from May 12 to May 25, 2011, before and after microstraining, fell within the range 0.029–0.116 µg·dm^−3^. In all 5 samples the MC-LR content did not exceed 1 µg·dm^−3^. The student’s t-test analysis showed significant differences between the amount of intracellular MC-LR (t_calc_ 2.40; t_crit_ 2.31 *p* > 0.05) before and after microstrainer filtration.

## 3. Discussion

To reduce the number of algae by approximately 80%, including cyanobacteria by 68%, in September 1994 microstrainers were installed at WTP Zielona Góra. The WTP is the only plant in Poland using microstrainers in the water-treatment system [2].

The use of the non-reagent process of particle filtration on microstrainers as a pre-treatment process for surface water abounding in algae is justified because it reduces the amount of precursors of oxidation by-products, improving physical and chemical indicators of water quality (e.g., turbidity, color, and total organic carbon (TOC)). Also, indirectly, reducing the amount of cyanobacteria causes a decrease in the amount of intracellular toxins [3].

The microstrainer filtration ensures considerable reduction in the content of phytoplankton, including cyanobacteria. In 2009, the effectiveness of micro-sieving in the removal of cyanobacteria (35.8–68.8%) increased along with their growing amount. The reduction in the amount of cyanobacteria was accompanied by decreased contents of the intracellular MC-LR [2]. In the last year of this study, with increased amounts of cyanobacteria, the dominant species was *P. agardhii*, which has high capacity for synthesizing hepatotoxins dangerous to humans. To eliminate any toxic threat from cyanobacteria to Zielona Góra water-supply system, one should introduce constant MC-LR monitoring in drinking water. In the last year of this research (2014), the maximum amount of cyanobacteria equaled 213 10^3^ org.·dm^−3^ (Table 2), which is one of the highest values in the research period. The dominant group was the *P. agardhii* species, which is particularly dangerous because it can produce up to three times as much microcystin as other cyanobacteria e.g., *Microcystis* sp. In Polish reservoirs, the most frequently occurring form of microcystin during *P. agardhii* domination is MC-RR. This process may have an adverse impact on the quality of treated water in the future. To estimate the cyanotoxic risk at a drinking-water intake, a systematic (annual) monitoring of cyanobacterial toxins should be introduced, especially in the summer–autumn season [18]. In the studies, cyanobacteria bloom comprising *P. agardhii* also occurred in September.

The analysis of microcystin amounts in May 2011 showed low content of intracellular MC-LR (<1 µg·dm^−3^), permitted in drinking water [19]. However, it must be stressed that in that period of research, the dominant species was *L. redeckei*, which is able to produce neurotoxins and not the studied hepatotoxins [20]. In the temperate zone, this species plays an important role in mesotrophic reservoirs, especially in colder seasons. Little is known about toxicity of this species, despite the urgency of the problem due to the frequent massive occurrences of this species in drinking-water reservoirs [21].

To eliminate any toxic threat from cyanobacteria to residents of Zielona Góra in drinking water supplied by waterworks, constant monitoring not only of hepatotoxins but also neurotoxins should be introduced. Microscopic analysis should be used as a preliminary test for estimating the potential cyanobacterial risk in water [22].

## 4. Materials and Methods

The efficiency of plankton removal on microstrainers at WTP Zawada was estimated by means of a hydrobiological analysis. 63 samples taken before and after microstraining were analyzed from May to September 2011–2014. The assumed sampling frequency was once a week on condition that the microstrainers were in operation. This research is aimed at estimating the levels of plankton removal by microstrainers at WTP and taking appropriate action (the cleaning of microstrainers, removing leaks).

The samples for hydrobiological analysis were taken using a Sedgewick-Rafter counting chamber with a volume of 0.001 dm^3^ in a given number of fields with the following parameters: height 1 mm, area 1 mm^2^. A MN 358/A (OPTA-TECH, Warszawa, Poland) microscope was used for observation [2]. For cyanobacteria with straight filaments, 100 µm was set as one individual (org.). Curved trichomes of *Dolichospermum* spp. and one colony of *Microcystis* spp. were indicated as individuals [23]. When converting the colony count from 0.001 dm^3^ into 1 dm^3^ sample concentration by the plankton net was taken into account [24]. The results of hydrobiological analysis were expressed as: org.·dm^−3^. Species identification was done with identification keys for algae [25,26,27,28].

To determine the microcystin concentration in water and cyanobacteria cells in May 2011, an additional series of chromatographic analyses (HPLC) of water samples was performed, which were then sent to the Department of Applied Ecology, University of Łódź. The chromatographic analysis was carried out with the use of Agilent 1100 chromatograph with Diode Array Detection (DAD) (Agilent, Waldbronn, Germany) [29].

A student’s *t*-test was used to analyze the significance of difference between the amount of plankton, including cyanobacteria and their toxins before and after microstraining at a significance level of α = 0.05. The statistical analysis was performed using the following programs: StatSoft Statistica version 10.0 and Microsoft Excel 2010.

## 5. Conclusions


The microstrainer removal of plankton and cyanobacteria from 2011 to 2014 was statistically significant. The quantity of removed plankton increased with increasing content in raw water.Cyanobacteria blooms occurred in surface water which is treated in WTP Zielona Góra. Cyanobacteria species occurring in WTP are capable of microcystin synthesis (e.g., *Planktothrix agardhii* bloom in September 2014). Therefore, cyanotoxin monitoring should be introduced especially in the summer–autumn season.The microstrainer process, by reducing the amount of cyanobacteria, contributes to a decrease in intracellular microcystins. However, after particle-filtration process, cyanotoxic cyanobacteria may still occur. Therefore, cyanotoxin testing is justified.The largest amount of cyanobacteria occurred in May and equaled 243·10^3^ org.·dm^−3^ (domination of *Limnothrix redeckei*) before the microstrainer process. To eliminate any toxic threat from cyanobacteria to residents of Zielona Góra in drinking water supplied by waterworks, constant monitoring of not only hepatotoxins but also neurotoxins should be introduced. It is necessary to perform microscopic analysis as a preliminary test for estimation of the potential cyanobacterial risk in surface water for drinking-water supply.


## Figures and Tables

**Figure 1 toxins-11-00285-f001:**
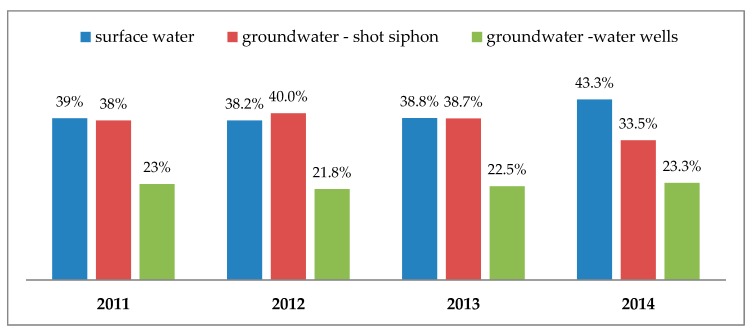
The percentage share of particular water sources in the consumption water supply for the inhabitants of Zielona Góra in the years 2011–2014.

**Figure 2 toxins-11-00285-f002:**
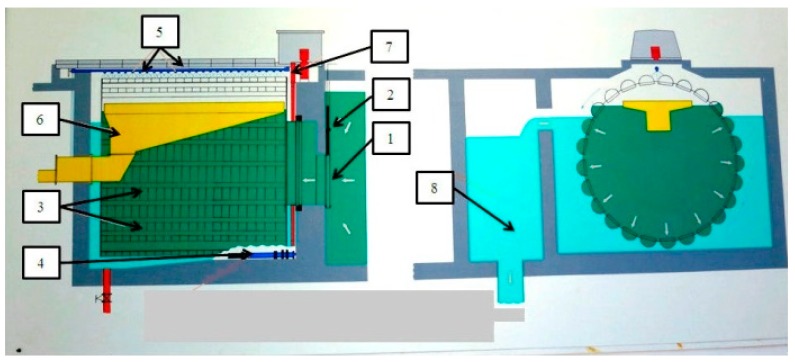
Microstrainers in WTP Zawada. **1** raw water supply; **2** slide flat; **3** mesh baskets; **4** submersible pump; **5** sprinklers; **6** drainage trough; **7** drum drive; **8** clean water outflow.

**Figure 3 toxins-11-00285-f003:**
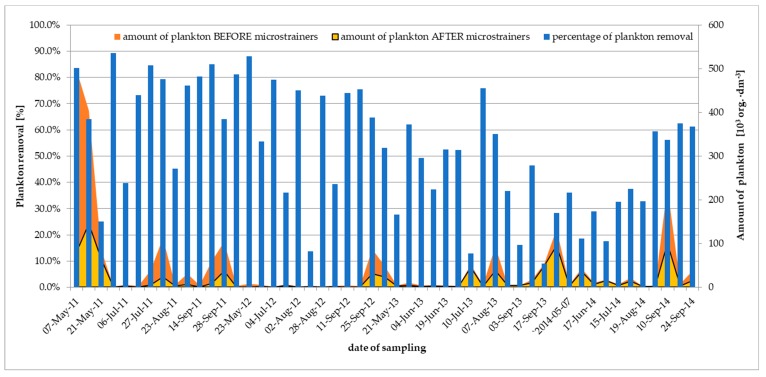
Efficiency of plankton removal after microstraining process in WTP Zawada from 2011 to 2014.

**Figure 4 toxins-11-00285-f004:**
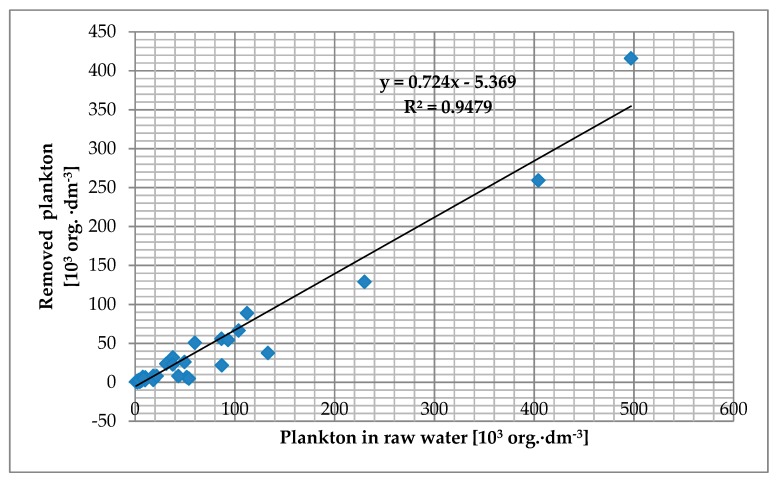
Relationship between amount of plankton in raw water and removed in WTP Zawada.

**Figure 5 toxins-11-00285-f005:**
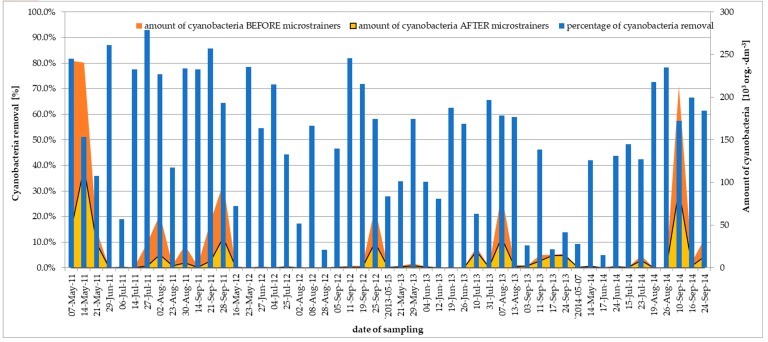
Efficiency of cyanobacteria removal after microstrainers process in WTP Zawada from 2011 to 2014.

**Table 1 toxins-11-00285-t001:** Microstrainers efficiency in removing plankton and cyanobacteria in WTP Zawada from 2011 to 2014.

Microorganisms	Number of Results	Range of Quantity [10^3^ org. dm^−3^]	Removal Efficiency	
			%	Average [%]
Plankton	8	> 100	47.4–83.7	65.1
	7	50–100	25.2–85.0	53.7
	7	10–50	13.6–76.8	53.7
	41	< 10	9.1–93.8	50.2
Cyanobacteria	4	> 100	51.1–81.1	65.9
	6	50–100	41.2–85.7	58.6
	10	10–50	13.9–93.0	49.7
	43	< 10	5.00–81.0	48.6

**Table 2 toxins-11-00285-t002:** Amount of cyanobacteria and intracellular MC-LR content before and after microstrainers filtration (WTP Zawada).

Date of Sampling [D/M/Y]	BEFORE Microstrainers Process		AFTER Microstrainers Process	
	Amount of Cyanobacteria Colony [10^3^ org.·dm^−3^]	Intracellular MC-LR [µg·dm^−3^]	Amount of Cyanobacteria Colony [10^3^ org.·dm^−3^]	Intracellular MC-LR [µg·dm^−3^]
11/05/2011	238	0.116	63.8	0.032
13/05/2011	241	0.098	118	0.040
18/05/2011	60.5	0.056	35.6	0.036
19/05/2011	43.9	0.044	28.2	0.032
25/05/2011	35.0	0.040	15.5	0.029

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
