# Peer review of "The Efficiency of Microstrainers Filtration in the Process of Removing Phytoplankton with Special Consideration of Cyanobacteria"

_toxins, 2019, doi:10.3390/toxins11050285_

Round 1
Reviewer 1 Report
The overall opinion about the manuscript is satisfactory, but as an applied study it can be of certain interest for practitioners.
The manuscript deals with filtration of plankton in surface water treatment. The study is built around the real case in Poland and focuses on monitoring of plankton for the assessment of cyanotoxic risk in drinking water supply.
The manuscript it not novel, but can be of some interest to practitioners, therefore adds a case study to the knowledge base.
The manuscript adheres to the journal’s standards.
Specific comments:
Sections 1.2 and 1.3 can be grouped under section Materials & methods.
Results presented in the Table 1 do not give a possibility to derive any analytical conclusions as efficiency ranges are wide (e.g. 5-99%).
Figure can gain better analytical value, if removal rate is expressed as %.
Table 2 is too difficult to read. It is suggested to move into annex or replace with another analytic material (graphic interpretation etc.).
Conclusion 1 could be better understood, if efficiency criteria are formulated in the introduction. It is not obvious that reduction of over 50% only in 60% samples is satisfactory. Proper risk assessment to substantiate this conclusion is missing in the paper.
Author Response
Response to Reviewer 1 Comments
Point 1: Sections 1.2 and 1.3 can be grouped under section Materials & methods.
Response 1: Sections 1.2 and 1.3 were grouped.
Point 2: Results presented in the Table 1 do not give a possibility to derive any analytical conclusions as efficiency ranges are wide (e.g. 5-99%).
Response 2: Table was redrafted and discussed again.
Point 3: Figure can gain better analytical value, if removal rate is expressed as %.
Response 3: I suppose it means figure 4, I tried to change expression but then the chart did not keep the proportions.
Point 4: Table 2 is too difficult to read. It is suggested to move into annex or replace with another analytic material (graphic interpretation etc.).
Response 4: Table 2 was moved into AppendixA.
Point 5: Conclusion 1 could be better understood, if efficiency criteria are formulated in the introduction. It is not obvious that reduction of over 50% only in 60% samples is satisfactory. Proper risk assessment to substantiate this conclusion is missing in the paper.
Response 5: Conclusion 1 was redrafted.
Reviewer 2 Report
There is some very important information present in this paper as the use of microstrainers is becoming increasingly popular. There are many elements in the use of English that should be improved in this paper. The lines on figures could also be made clearer and overlapping information fixed.

Author Response

(The authors gave the same response as above.)
